# Exploring the Formation Mechanism, Evolution Law, and Precise Composition Control of Interstitial Oxygen in Body-Centered Cubic Mo

**Hai-Rui Xing** [1,2], **Ping Hu** [1,2,*], **Chao-Jun He** [1,2], **Xiang-Yang Zhang** [1,2], **Fan Yang** [1,2], **Jia-Yu Han** [1,2], **Song-Wei Ge** [1,2], **Xing-Jiang Hua** [1,2], **Wen Zhang** [3,*], **Kuai-She Wang** [1,2] and **Alex A. Volinsky** [4]

[1] School of Metallurgy Engineering, Xi'an University of Architecture and Technology, Xi'an 710055, China
[2] National and Local Joint Engineering Research Center for Functional Materials Processing, Xi'an University of Architecture and Technology, Xi'an 710055, China
[3] Northwest Institute for Non-Ferrous Metal Research, Xi'an 710016, China
[4] Department of Mechanical Engineering, University of South Florida, 4202 E. Fowler Ave. ENG 030, Tampa, FL 33620, USA
* Correspondence: huping@xauat.edu.cn (P.H.); gwenzh@163.com (W.Z.); Tel./Fax: +86-158-0296-8790 (P.H.); +86-029-86234723 (W.Z.)

**Abstract:** Interstitial oxygen (O) on the formation mechanism and enrichment distribution of body-centered cubic (BCC) molybdenum (Mo) has rarely been reported, and the O usually can cause serious brittle fracture in Mo. In this paper, we studied the formation mechanism and evolution of oxygen (O) when it was precisely controlled in the range of 3700–8600 parts per million (wppm). It was found that, with an increase in O concentration, O element not only existed in the form of solid solution but generated O element with different valence states in Mo metal. Large amounts of $MoO_2$, $MoO_3$, and $Mo_4O_{11}$ intermediate oxides were identified by electron probe micro-analyzer (EPMA) and X-ray photoelectron spectroscopy (XPS). Thermodynamic calculations revealed the formation process of oxides, and authenticity of the presence of O was verified by XPS. Enrichment and distribution of O element were analyzed by scanning electron microscopy (SEM), energy dispersive spectroscopy (EDS), and EPMA. Moreover, the compressive yield strength and hardness of Mo were greatly affected by O content range of 4500–8600 wppm. Our study is helpful to understand the behavior of interstitial impurity O in refractory Mo metals and provides important guidance for development of high-purity rare Mo metals.

**Keywords:** molybdenum; interstitial oxygen; formation mechanism; evolution; precise control; powder metallurgy; microstructural evolution

## 1. Introduction

Molybdenum (Mo) has been widely used in aerospace engineering, military, and electronics industries because of its excellent performance, including high-temperature strength, creep resistance, thermal conductivity, corrosion resistance, and low thermal expansion coefficient [1–10]. With development of high-tech materials, such as modern electronics, semiconductors, and high-quality thin-film materials, the requirements for the purity of Mo are increasing [11]. However, non-intrinsic brittleness problems, such as extrinsic brittleness of interstitial C, O, and N elements at GBs (GBs), will lead to brittle fracture of Mo alloys [12–15], thus greatly reducing performance [16–22].

There are two reasons for brittleness in Mo alloys: intrinsic and non-intrinsic. Intrinsic brittleness is caused by transition between the properties of covalent bonds distributed in the d electron shell and performance of metallic bonds reflected in the outermost shell electrons [23].

Segregation of some interstitial impurities (C, O, and N) at the GBs is considered to be the main reason for Mo grain boundary embrittlement and low toughness [24–26].

In this sense, mastering and controlling the content of O in Mo will effectively improve strength and ductility [27–33]. Zhang et al. [34] have adjusted the concentration of O solutes in Vanadium (V). The results show that yield strength, strain hardening rate, and ultimate tensile strength of V are simultaneously improved to achieve strengthening or embrittlement effects. It also provides insights into design of high-performance refractory metals utilizing O solutes. However, there is little research on the formation mechanism and evolution law of O solutes in refractory metal [35,36].

Compared with other refractory metals, the solubility of element O in Mo is low, and its mass fraction does not exceed 0.0002% mass fraction at room temperature, which results in chemical analysis of element O in Mo being difficult. Currently, rapid development of atomic probe tomography (APT) provides a new method for detecting non-metallic elements in Mo [37]. Waugh and Southon [38,39] carried out early atomic probe experiments and confirmed the existence of elemental O at GBs. Three-dimensional atomic probe microscopy (APM) shows that grain boundary segregation of elemental O and N promotes intergranular fracture of these intrinsically brittle GBs [40], which has also been proven by first-principles calculations [41,42]. C segregation reduces the local O content, which can enhance the Mo grain boundary [43,44]. Atom probe studies have shown the presence of elemental O and N at single GBs by powder metallurgy (PM) [45,46]. Transmission chrysanthemum pond diffraction (TKD) and APM [47] are used to investigate the different high-angle GBs in Mo, indicating that trace interstitial elements O and N are separated from high-angle GBs [48]. The detection methods of trace elements include atomic emission spectroscopy (AES) [49], atomic absorption spectrometry (AAS) [50], inductively coupled plasma atomic emission spectrometry (ICP-AES) [51], inductively coupled plasma mass spectrometry (ICP-MS) [52], etc. Detection of trace interstitial element O in Mo metal is studied by gas extraction method to analyze O contents at the ppm level [53]. Compared with previous reports [54–56], this study has explored the formation mechanism, evolution law, and precise composition control of O in Mo metal during powder metallurgy preparation, which has never been reported before.

In this work, we proposed the powder metallurgy approach to introduce O by doping $MoO_2$ powder to improve the accuracy and operability of O detection results. It can accurately control the O content and solve the difficulty to accurately characterize the trace impurities in Mo metal. The content of O in sintered Mo and its influence on the microstructure and composition were analyzed by SEM, EPMA, XPS, and electron back scattering diffraction (EBSD). The formation mechanism, evolution law, and precise composition control of molybdenum were further discussed. Then, room temperature compression was used to study the strength change in Mo after doping elemental O. Finally, the effects of elemental O on GBs were explored. These results are expected to increase understanding of the behavior of interstitial impurity O in powder metallurgy of refractory Mo metals. It is also of great significance to study element detection and powder technology of the Mo-O system.

## 2. Experimental

Initially, the 99.95% purity Mo powder (FMo-1, ~3 μm, containing 1700 ppm O, Jinduicheng Molybdenum Co., Ltd. Xi'an, China) and $MoO_2$ (~5 μm, Jinduicheng Molybdenum Co., Ltd. Xi'an, China) were used as initial materials. In this study, we proposed powder metallurgy as a solution for adding elemental O, where the raw materials were a solid–solid mixture of Mo and oxide powders. In this study, we proposed powder metallurgy as a solution to add element O, in which the raw material was a solid–solid mixture of Mo and oxide powders. The content of O can be increased in the range of 3700–8600 parts weight/million (wppm).

### 2.1. Materials and Processing

The irregular morphology of pre-mixed Mo powder, the layered morphology of $MoO_2$ powder, and the distribution of mixed powders appear in Figure 1a–c. The chemical composition of Mo with different contents was listed in Table 1. The SEM images showed that the oxide flake morphology in the mixed Mo powder can be uniformly distributed.

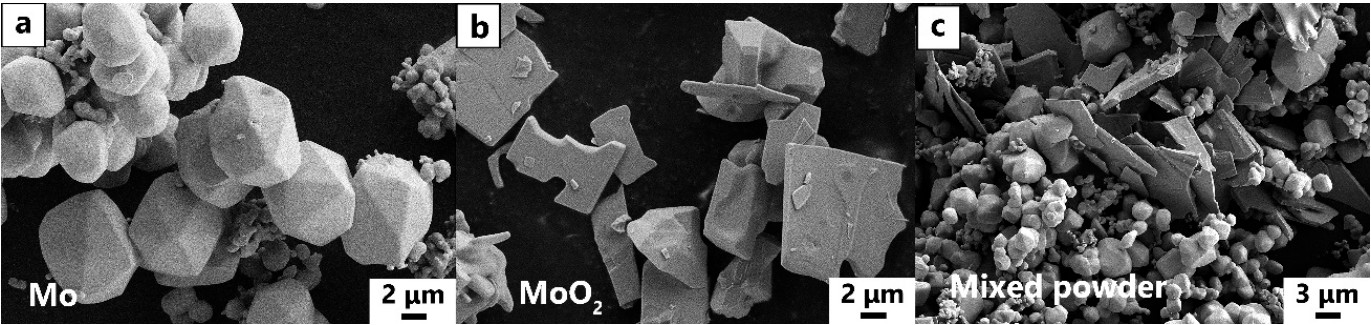

**Figure 1.** Morphology of (**a**) pure Mo, (**b**) $MoO_2$ powders, and (**c**) Mo and $MoO_2$ powders mixture.

**Table 1.** Designed chemical composition of Mo in wt.%.

| Number | O | $MoO_2$ | Mo |
|---|---|---|---|
| 1 | - | - | Bal. |
| 2 | 0.6 | 4.92 | Bal. |
| 3 | 0.8 | 6.56 | Bal. |
| 4 | 1 | 8.2 | Bal. |
| 5 | 2 | 16.4 | Bal. |

Mo specimens with different O contents were prepared by powder metallurgy as follows, as shown in Figure 2. (i) In a three-dimensional mixer (SBH-5, China, 22 rpm rotational speed, Shanghai Tianhe Machinery Equipment Co., Ltd., Shanghai, China), $MoO_2$ and Mo powders with different contents were mixed in air for 90 min by solid–solid mixture method. The 4.92 wt.%, 6.56 wt.%, 8.2 wt.%, and 16.4 wt.% $MoO_2$ powders were fully mixed with 200 g Mo powders (Table 1). (ii) The powders were milled by planetary ball mill (QM-3SP2, Nanjing Chishun Technology Development Co., Ltd., Nanjing, China) for 30 min. The ratio of ball powder was 1:1, and the rotating speed was 200 rpm. The impurity C atom would be introduced during ball milling. The size of the prepared cylindrical sample was $\Phi 60 \times 10$ mm. In the pressing process, sodium stearate lubricant was used to reduce the friction between powder particles and die, and increased the fluidity of particles. (iii) The pressed samples were isothermally sintered at 1800 °C in a vacuum tube furnace (ZT-40-20, Xiamen Dikun Technology Co., Ltd., Xiamen, China). Isothermal sintering was carried out at 800 °C, 1200 °C, 1600 °C, and 1800 °C for 2 h, 1 h, 2 h, and 1 h, respectively, as shown in Figure 3. The temperature gradually rose to the sintering temperature and was kept constant during continuous isothermal sintering. The isothermal sintering method can accurately control the temperature and improve the sintering heating efficiency. In the process of vacuum sintering, introduction of C mainly came from the ball milling and pressing in the powder metallurgy.

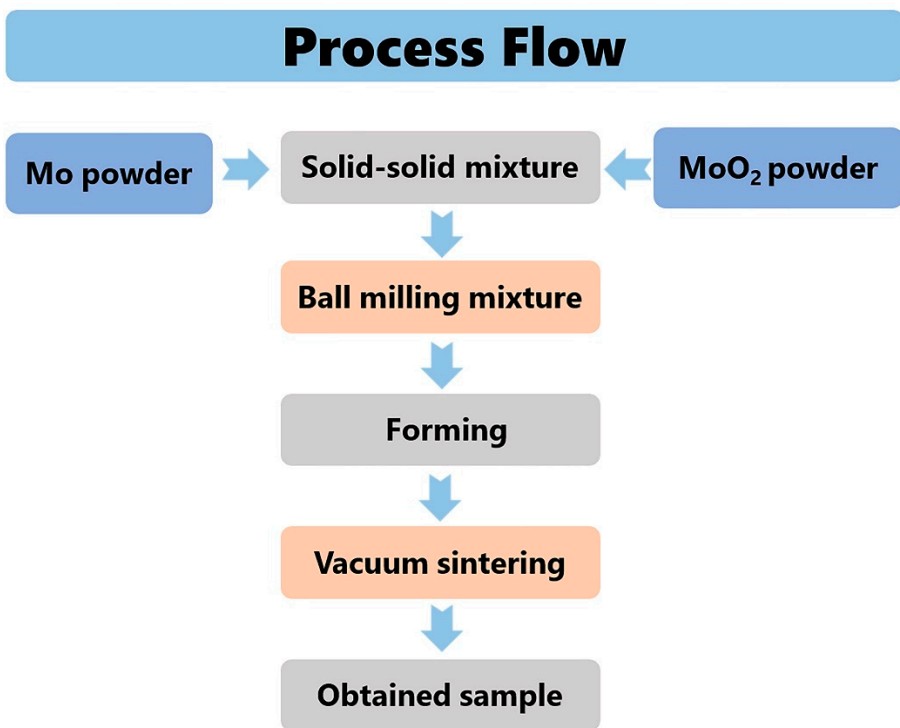

**Figure 2.** Powder metallurgy process flow.

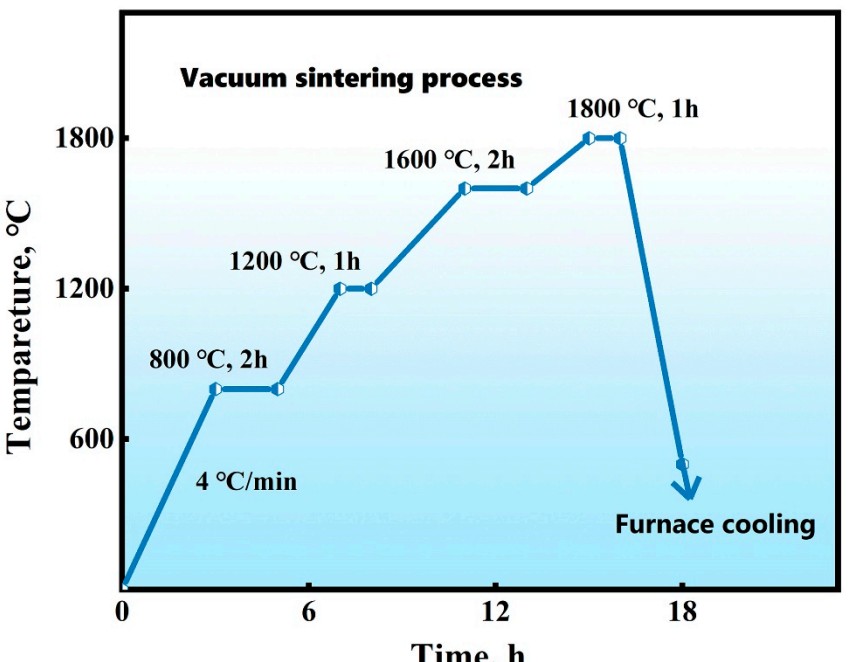

**Figure 3.** Vacuum sintering process.

### 2.2. Characterization

O and C contents were tested by oxygen–nitrogen analyzer (RO-316, Steel Research NAK Testing Technology Co., Ltd., Beijing, China) and carbon–sulfur analyzer (CS-344, Steel Research NAK Testing Technology Co., Ltd., Beijing, China). The sintered samples were pulverized and the freshly fractured microstructure were observed. The powder samples and fracture morphology were examined by a scanning electron microscope (Gemini SEM 500, Carl Zeiss Management Co., Ltd., Shanghai, China) equipped with EDS (Oxford UltimMax100 energy spectrum detector). For the main element Mo (>20% wt.%), the allowable relative error was $\leq\pm5\%$. For the O content within $\pm$ 0.5% wt.%~1% wt.%, the allowable relative error was $\leq 50\%$. Therefore, SEM can only conduct qualitative and semi-quantitative composition analysis of oxygen element. All sintered Mo samples were phase-analyzed using Cu-K$\alpha$ radiation of X-ray diffraction (XRD) spectra at room temperature. The EPMA (JXA-8230, JEOL Science and Trade Co., Ltd., Beijing, China) was used to analyze elements' distribution and content. Thermo Scientific K-Alpha XPS (Thermo Fisher Scientific, Shanghai, China) was used to examine the elemental distribution at the GBs. The radiation source was Al-K$\alpha$ with 400 μm beam spot and 0.1 units testing step. The standard binding energy of C 1 s was 284.8 eV. The Instron 5969 universal testing machine (Instron, shanghai, China) was used for testing $\Phi$5 mm $\times$ 9 mm samples in room temperature compression tests. The strain rate was 0.001 mm/s. The microhardness experiments of samples were tested at 200 g load and 10 s residence time.

### 3. Results and Discussion

### 3.1. Oxygen Content and Phase Composition of Mo

First, the mixed powder and sintered samples were tested by an oxygen–nitrogen analyzer (RO-316, Steel Research NAK Testing Technology Co., Ltd., Beijing, China), and the results were listed in Table 2. The origin of O can be divided into free O in Mo powder, including O adsorbed on the surface of Mo powder, O in the deep layer of Mo powder, and introduced $MoO_2$ oxides [57]. Composition analysis showed that the mixed powder and vacuum-sintered samples contained O levels of 1700–20,000 weight parts per million (wppm, red line in Figure 4) and 38,700–8600 wppm (blue line in Figure 4). Here, the sintered samples were named O-1 to O-6. The O-1 sample was the control without adding $MoO_2$. The O content measured from O-2 to O-6 samples was 3700 wppm. The O content of 0–16.4 wt.% $MoO_2$ increased from 3700 wppm to 8600 wppm. By comparing the powder and sintered samples, it was found that the sintering deoxidation reaction greatly reduced the O content.

**Table 2.** O content in preparation of Mo specimens (ppm by weight, wppm).

| Number | | Powder Samples | Sintered Samples |
|---|---|---|---|
| 1 | O-1 | 1700 | 3700 |
| 2 | O-2 | 6000 | 4500 |
| 3 | O-3 | 8000 | 4700 |
| 4 | O-4 | 10,000 | 6200 |
| 5 | O-5 | 20,000 | 8600 |

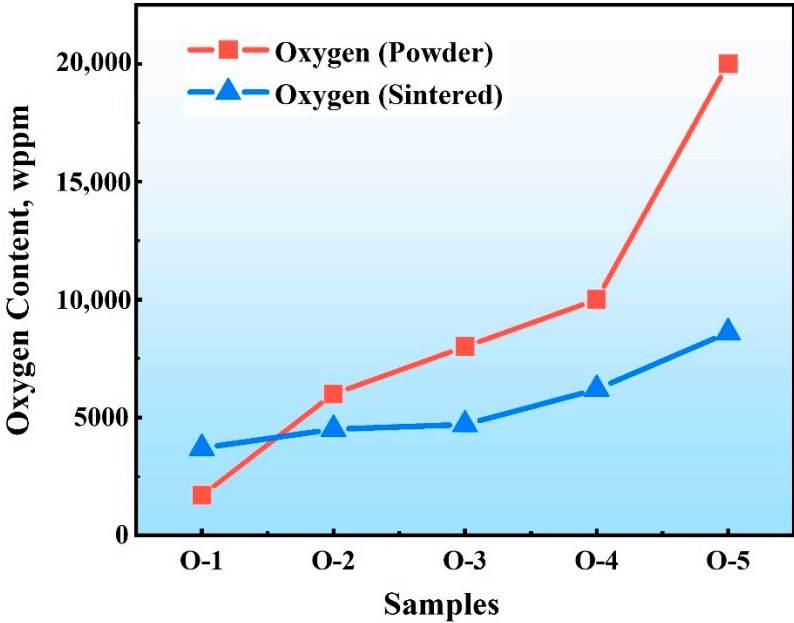

**Figure 4.** O content in powder and sintered samples during powder metallurgy preparation.

Figure 5 showed the XRD results [58] of the O-1-to-O-5-sintered Mo samples. From the XRD pattern (Figure 5a), the reflections obtained were only from the BCC Mo (JCPDS 89-4896), corresponding to (1 1 1), (2 0 0), (2 1 1), and (2 2 0) Mo crystal planes. No new reflections of $MoO_2$ were found in the O-1-to-O-4-sintered samples, mainly because the oxide content in these samples was very low. In addition to X-ray reflection of the Mo metal phase, there were also (1 1 1), (2 1 1), (2 2 2), and (2 1 3) crystal plane reflections. This indicated that the bcc structure of the single-phase Mo metal did not change when the O content was 8200 wppm. The peak value of $MoO_2$ was detected in the O-5 sample, indicating that addition of 16.4 wt.% O could cause Mo metal to contain $MoO_2$ phase. The variation in lattice parameters was related to the elements' solid solution determined by lattice distortion. As we can see in Figure 5b, the addition of O caused the 2θ increasing in the measurements [59]. From the relationship between the lattice constant and θ ($a = \frac{\lambda\sqrt{2}}{2}\frac{1}{\sin\theta}$), we can obtain that the lattice constant changed. The lattice parameter results were calculated from X-ray measurements combined with full spectrum fitting refinement. As shown in Figure 5c, the results of lattice parameters were obtained by X-ray measurement combined with full spectrum fitting and refinement. The lattice parameters of five samples increased with an increase in O content. The element O was mainly presented in the Mo matrix in the form of solid solution, which would distort the Mo lattice.

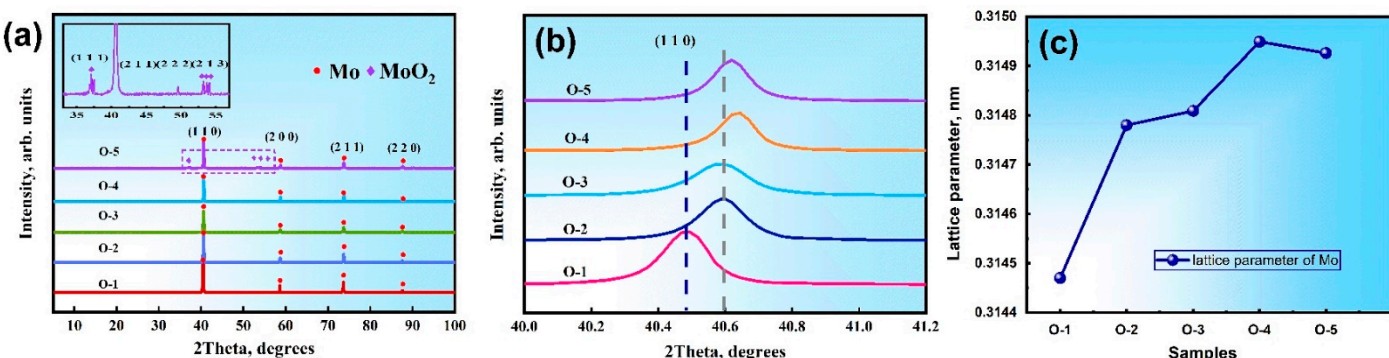

**Figure 5.** XRD patterns of Mo samples. (**a**) The overall XRD curves of Mo. (**b**) The enlarged (110) peaks show the shift of the peaks. (**c**) The lattice parameters of Mo.

### 3.2. Oxygen Effects on Mo Microstructure and Composition

The fracture surfaces of O-1 to O-5 specimens sintered at 1800 °C were shown in Figure 6. The upper right corner marked the O content of the sintered samples. The sintered pores formed by volume diffusion were marked with a circle. A partially enlarged view of the fracture surfaces is shown in Figure 6d,f. Figure 6a shows that the small pores in the O-1 sample were caused by volatilization and reduction of gas or impurities and release of a large amount of gas during sintering. Pore shrinkage required a material migration mechanism caused by volume diffusion. As shown in Figure 6b, when the O-2 sample containing 4500 wppm O was sintered in a vacuum, pores appeared on the fracture surface, the grains began to grow, and the fracture was intergranular. For O-4 sample containing 6200 wppm O, it was found in the enlarged view (Figure 6c,d) that the width at the GBs was about 5 μm. We can see that the fracture mode in Figure 6c was consistent with the O-1 and O-2 samples, but the morphology at the GBs was different. As shown in Figure 6d, it was found that different structures begin to appear at the GBs by SEM with magnification of five times, which was proved to be formation of oxides later. Similarly, precipitates were found at the GBs of the 8600 wppm O-5-sintered sample in Figure 6e,f, indicating that the precipitates appeared at the GBs. When the O content was different, the fracture morphology did not change. Due to doping of O, the grain boundary fracture was intergranular. The increase in O content changed the porosity and grain size of the sintered Mo samples but did not change the intergranular nature of fracture of the sintered Mo samples' fracture. It can be seen from the fracture morphology of Mo with different O content that there was a precipitation phase network distributed at the GBs, which affected the microstructure of Mo. In addition, the precipitates also change the structure and morphology of molybdenum GBs and affect the microstructure of molybdenum. To further explored the element composition of the precipitate formed at the GBs of O-4 and O-5 samples, we would use EDS to detect the element content for qualitative judgment.

SEM images and EDS analysis results of the O-1-sintered sample were shown in Figure 7a. The Mo content in spectrum 1 was 100 at.%. Figure 7b shows sintered Mo sample O-2 with 4500 wppm O. It can be seen from spectrum 2 that it is mainly Mo on the GBs of the O-2-sintered sample. The EDS results show that the O-2 sample was enriched with Mo and O elements at the GBs on the fracture surface in Figure 7b. The atomic ratio of these elements was 34.64:65.36, which was approximately equal to 1:2. Therefore, the grain boundary product can be preliminarily identified as $MoO_2$. Further verification and its formation mechanism were presented in the following sections. From the EDS surface scanning distribution in Figure 7c–f, the elemental O was uniformly distributed in the GBs and within the grains [60]. The bright red area indicated that the elemental O was preferably enriched at the GBs. It can be clearly seen that the sediment presented an irregular network distribution at GBs in Figure 7c,d. According to the total map, the atomic ratio of Mo to O is 59.92:40.08. In summary, O preferred to segregate at GBs with an increase in O content, Mo oxides appeared in some grain boundary areas of the Mo fracture surface, and the oxides were distributed in the grain boundary areas, similar to the network structure. The O-3-to-O-5-sintered Mo samples were also analyzed by EDS spectroscopy, and it was found that the O content at the GBs gradually increased in Figure 8a–e. In particular, the atomic ratio of Mo and O at the GBs of O-5 sample is 29.05:70.9 (Figure 8e,f). Through analysis of GBs by EDS, we can find that the O content was gradually increasing in the GBs region. Combined with Figure 7c–f, it can be seen that the GB region of the Mo fracture mainly contained O element, and the precipitates formed by O element were distributed on the fracture surface in a network. To further discuss the distribution and content of O, C, and Mo elements in different sintered Mo samples, the surface analysis of Mo was analyzed by EPMA.

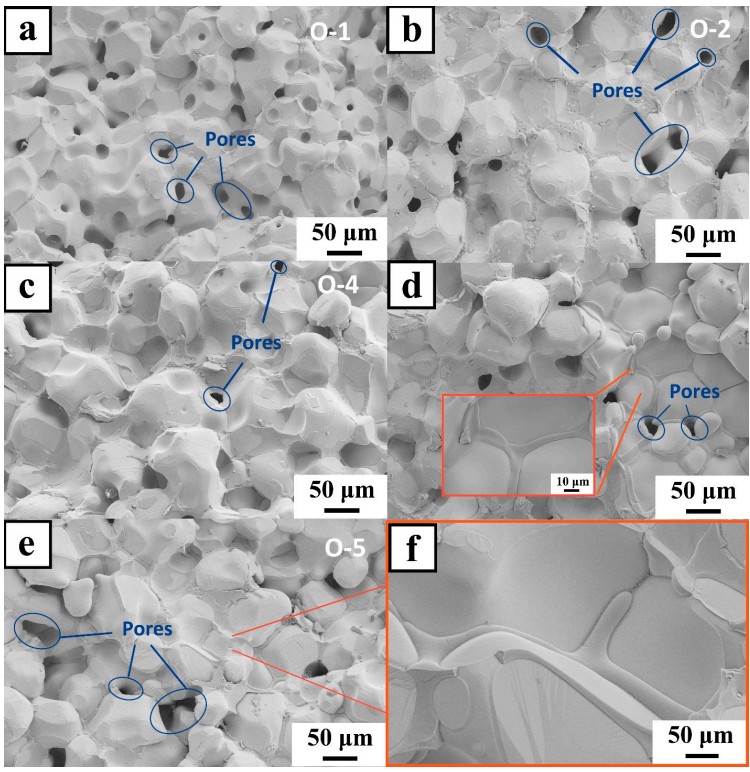

**Figure 6.** The fracture surfaces of Mo-sintered samples: (**a**) O-1 sample with 3700 wppm O; (**b**) O-2 sample with 4500 wppm O; (**c–f**) O-4 and O-5 samples with 6200 wppm and 8600 wppm O, respectively. The circles outline the sintered pores in Mo.

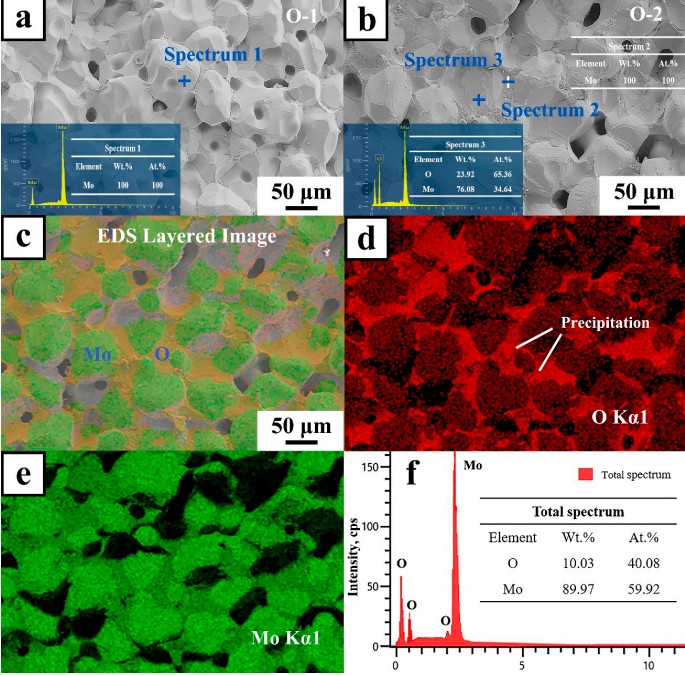

**Figure 7.** SEM images and EDS spectra of O-1 and O-2 samples during preparation: (**a**) O-1; (**b**) O-2; (**c–f**) EDS mapping analysis of the O-2 sample.

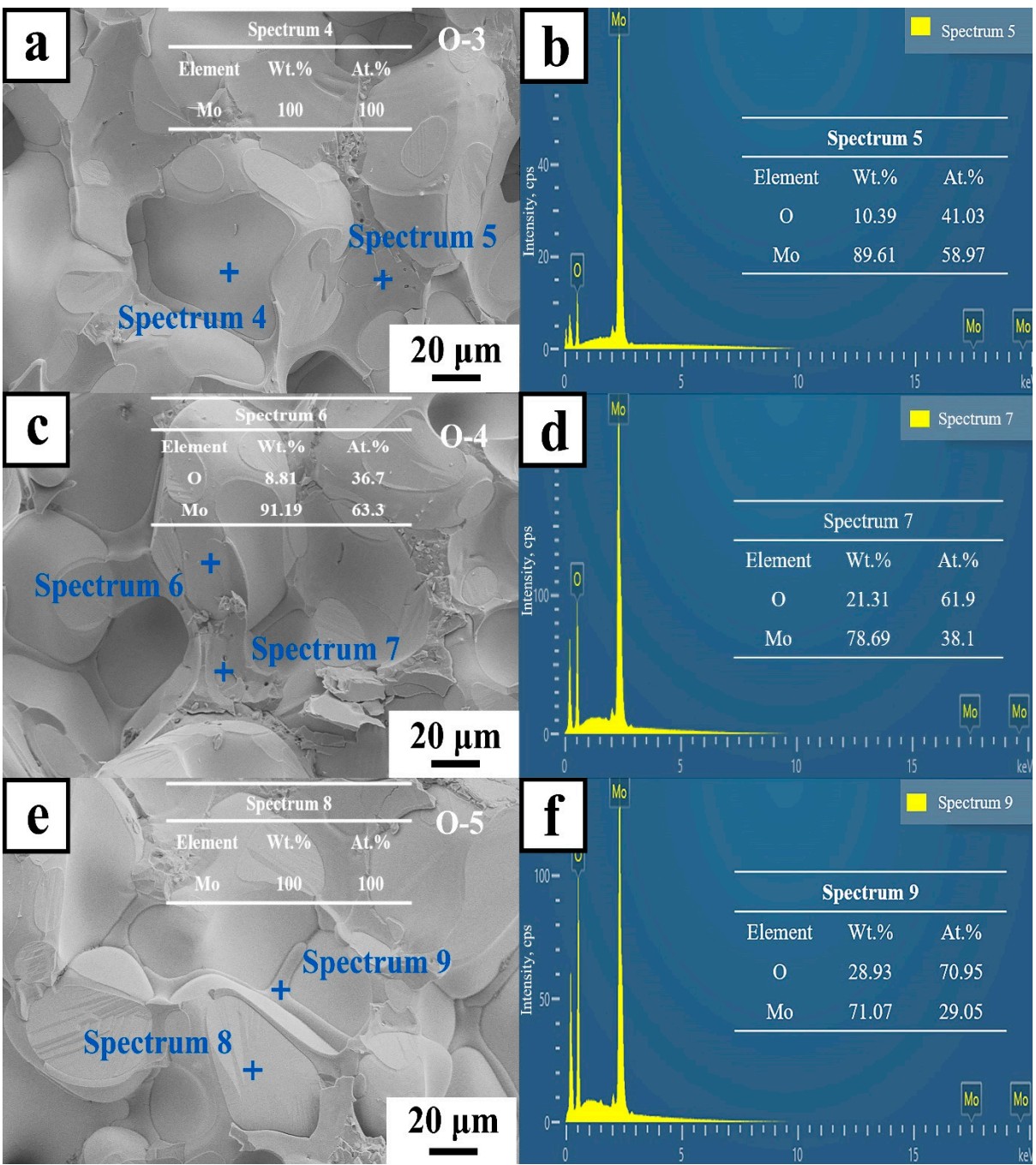

**Figure 8.** SEM images and EDS spectrum of O-3 to O-5 samples during preparation: (**a**,**b**) O-3 sample with 4700 wppm O; (**c**,**d**) O-4 sample with 6200 wppm O; (**e**,**f**) O-5 sample with 8600 wppm O.

The backscatter images of O-1 to O-5 samples in this region and the EPMA element distributions of Mo, O, and C were shown in Figure 9a–d. In the EPMA map, the color depth represented the element content. The higher the content, the stronger the signal. The white boxes represented the grain boundary regions of the Mo matrix. From the backscatter image and element distribution in Figure 9a, when the O content was 3700 wppm, the surface color of the O-1-sintered sample was uniform, indicating that O and C were almost uniformly distributed. When the O content increased to 4500 wppm in Figure 9b, the O in the O-2-sintered sample began to accumulate at the GBs. The color became more obvious and brighter, but there was no obvious accumulation of C element, which is consistent with Figure 6b in the SEM analysis. Element O can initially exist in the form of solid solution.

With an increase in O content, the color of Mo changed significantly (Figure 9c). We can see that the O element will exist at the GBs in the form of oxide. The EPMA maps analysis showed that the O element was segregated at most of the GBs in Mo, which could more clearly and accurately detect the distribution information of the O element, and the Mo samples showed that a similar network structure was formed. The carbon content was relatively small and mainly distributed in Mo pores. When the O content increased to more than 6200 wppm, the pores on the surface of Mo metal increased obviously. On the one hand, O could react with C content during vacuum sintering, releasing a great deal of gas. On the other hand, O would affect densification of the sintering process. Therefore, these had an impact on formation of pore morphology.

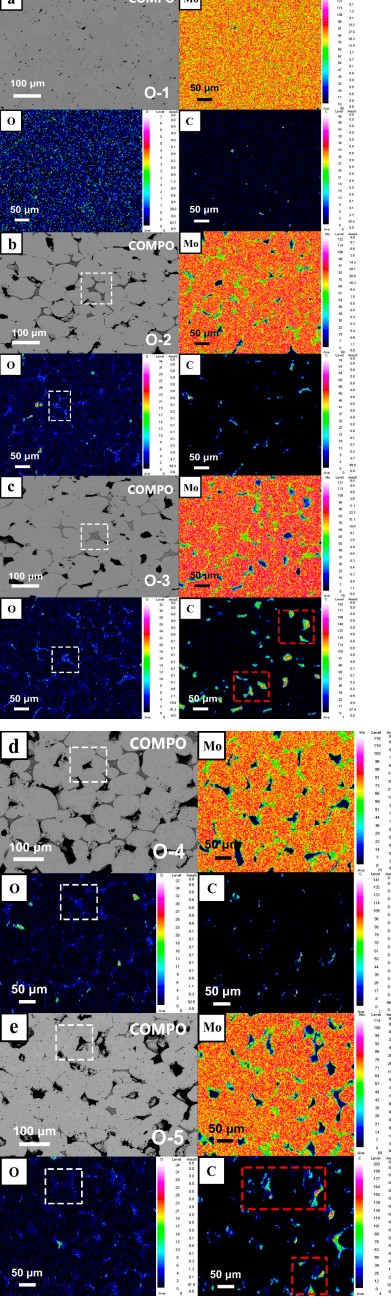

**Figure 9.** EPMA maps of O-1 to O-5 samples: (**a–e**) composite image and the specific Mo, O, and C elements maps. The white box represents segregation of elemental O in the grain boundary region of the Mo matrix. The red box indicates segregation of the C element in the pores of the Mo matrix.

When the O content was 8600 wppm, the distribution of the O element was more than in Figure 9b, and the C element had obvious segregation at the pores (indicated in the red box). Similarly, as seen in Figure 9e, the O-4 sample with 6200 wppm O also had more pores. The segregation of elemental O at the grain boundary was obvious, and the bright area of O was widely distributed. This indicated that the impurity C element reacted directly or indirectly with O in some pores, reducing the O content in the pores and the overall O content in sintered samples. In a word, the EPMA scanning results showed that the color of O changed significantly, revealing that the elemental O was mainly distributed at GBs, which could accurately describe the distribution of O in Mo, while the lower C content was mainly distributed in the pores of Mo.

The EPMA was further used to measure the O element in samples O-1 to O-5 [61]. The measurements' positions were shown in Figure 10, and the results were listed in Table 3. Since the Mo standard was tested before the EPMA test, there was a test error for this standard. Therefore, the total element content of all the samples in Table 3 exceeds 100 wt.%. With the addition of O, the average contents of O-1 to O-5 samples gradually increased in Mo, which were 0.4 wt.%, 12.86 wt.%, 15.64 wt.%, 15.31 wt.%, and 16.26 wt.%, respectively. The O content of the O-5 sample was significantly higher than the O-1 sample, which was consistent with the above results. It was found by electron probe scanning that the O content at positions 1, 2, and 3 in Figure 10a remained unchanged. In Figure 10b, the mass percentage of O at position 2 was 15 times that at position 1, indicating that the content of O at the GBs at position 2 was much higher than that inside the grain. The mass percentage of O in positions 1, 3, and 5 (grain boundaries) of O-3 samples in Table 3 was about 40 times higher than that in positions 2 and 4 (within grains). The mass percentage of O in positions 1, 3, and 5 (grain boundary) of the O-3 sample (Table 3) was about 40 times higher than that in positions 2 and 4 (intragranular). Therefore, it was quantitatively proven that the O added in the sintering process was segregated in the GBs of the Mo matrix in addition to the reaction with C. The average content of the total O element in the GBs was about 12.09 wt.%. In this paper, the content of the O element up to 8600 wppm can be introduced by the solid doping process of adding Mo dioxide powder, which realized enrichment of O at the grain boundary and solid solution of O and the Mo matrix. In this paper, through the solid doping process of adding Mo dioxide powders, content of O element as high as 8600 wppm can be introduced, realizing the enrichment of O at the GBs and solid solution of O and Mo matrix.

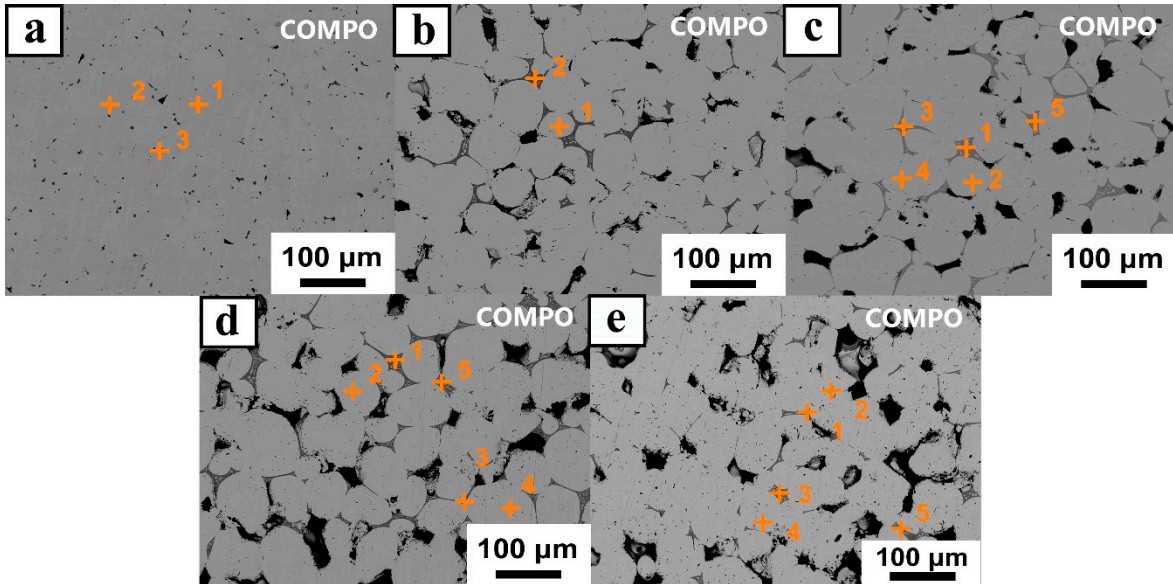

**Figure 10.** Schematic diagram of positions for EPMA measurements of (**a**) O-1; (**b**) O-2; (**c**) O-3; (**d**) O-4; (**e**) O-5.

**Table 3.** Elements' content measured in Figure 10a–e.

| Position | Test No. | Element (wt.%) | | | |
|---|---|---|---|---|---|
| | | **C** | **O** | **Mo** | **Total** |
| O-1 | 1 | 0.281 | 0.49 | 101.26 | 102.04 |
| | 2 | 0.335 | 0.561 | 101.95 | 102.85 |
| | 3 | 0.497 | 0.155 | 101.91 | 102.56 |
| | Average | 0.371 | 0.4 | 101.53 | 102.3 |
| O-2 | 1 | 0.747 | 1.728 | 102.11 | 104.59 |
| | 2 | 3.175 | 23.99 | 76.718 | 103.89 |
| | Average | 1.961 | 12.86 | 89.415 | 104.24 |
| O-3 | 1 | 0.269 | 25.65 | 76.314 | 102.23 |
| | 2 | 0.739 | 0.643 | 103.17 | 102.68 |
| | 3 | 0.689 | 25.95 | 75.927 | 102.57 |
| | 4 | 0.885 | 0.663 | 101.4 | 102.97 |
| | 5 | 0.682 | 25.28 | 76.013 | 101.97 |
| | Average | 0.653 | 15.64 | 86.565 | 102.48 |
| O-4 | 1 | 0.207 | 24.40 | 76.146 | 100.75 |
| | 2 | 0.302 | 1.115 | 100.93 | 102.35 |
| | 3 | 0.663 | 25.386 | 76.931 | 102.98 |
| | 4 | 0.745 | 1.061 | 103.13 | 104.94 |
| | 5 | 0.688 | 24.57 | 76.343 | 101.6 |
| | Average | 0.521 | 15.31 | 86.697 | 102.52 |
| O-5 | 1 | 0.629 | 26.43 | 77.11 | 104.17 |
| | 2 | 0.979 | 1.241 | 101.48 | 103.7 |
| | 3 | 0.403 | 26.08 | 76.390 | 102.87 |
| | 4 | 0.571 | 1.042 | 99.939 | 101.55 |
| | 5 | 0.23 | 26.52 | 75.8 | 102.55 |
| | Average | 0.562 | 16.26 | 86.144 | 102.97 |

*3.3. Oxygen Effects on Mo Mechanical Performance*

To explore the effects of O content on the mechanical properties of Mo-sintered specimens, room temperature uniaxial compression and hardness tests were performed. The compressive strength, yield strength, Vickers hardness, and grain sizes of the five groups of samples were obtained, as listed in Table 4. For O-1 sample of 3700 wppm, there was no obvious yield point at room temperature compression, which broke at 0.59 strain and 1384.85 MPa compressive stress. When the O content was 4500 wppm, the compressive strength of the O-2 sample dropped sharply to 98.83 MPa. When the O content was increased to 8600 wppm, the minimum tensile strength was 7.84 MPa, which was almost 99.43% lower than the O-1 sample in Figure 11a. The yield strength and hardness of O-2 to O-5 samples were shown in Figure 11b. The O-1 sample was deformed and the compression engineering stress–strain curve showed an upward trend. No yield strength data were obtained, indicating that the O-1 sample had no yield point at a strain rate of 0.001 s$^{-1}$. The compressive yield strength of O-2, O-3, and O-4 samples was 323.95 MPa, 295.8 MPa, and 341.74 MPa, respectively. The yield strength of the three samples decreased slightly. The yield strength of the 8600 wppm O-5 sample was about 43.4% lower than the O-4 sample of 6200 wppm. Notably, with an increase in O content, not only the microstructure of Mo had been changed but its compressive properties were also greatly affected, apparently decreasing the room temperature compressive properties of Mo.

**Table 4.** Mechanical properties and grain size of sintered Mo specimens.

| Number | O Content wppm | Compressive Strength MPa | Yield Strength MPa | Vickers Hardness HV | Grain Size μm |
|--------|----------------|--------------------------|--------------------|---------------------|---------------|
| O-1 | 3700 | 1384.85 | - | 125.86 ± 8.28 | 17.35 |
| O-2 | 4500 | 98.83 | 323.95 | 151.92 ± 13.79 | 18.36 |
| O-3 | 4700 | 35.97 | 295.8 | 143.81 ± 9.01 | 20.11 |
| O-4 | 6200 | 12.43 | 341.74 | 147.16 ± 13.42 | 21.53 |
| O-5 | 8600 | 7.84 | 193.58 | 136.01 ± 9.88 | 24.7 |

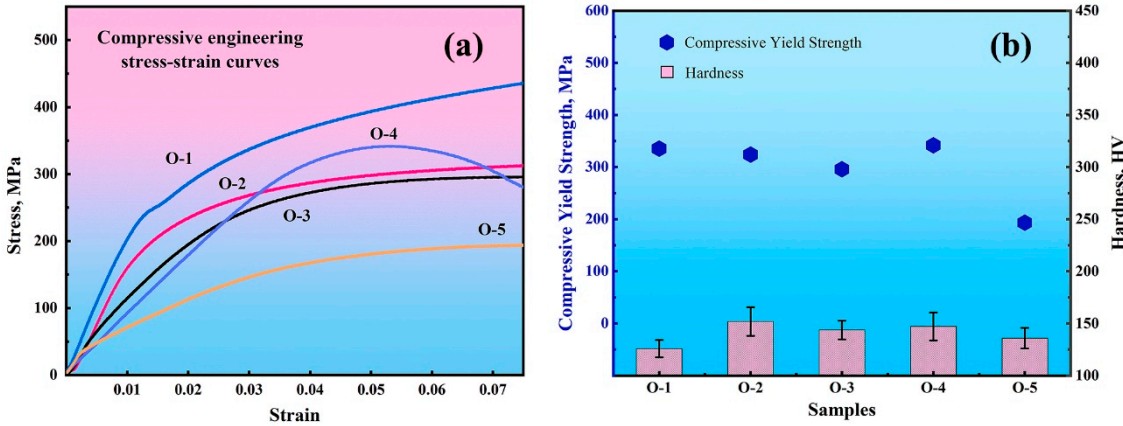

**Figure 11.** (**a**) Compressive engineering stress–strain curves of O-1 to O-5 samples deformed at 0.001 s$^{-1}$ strain rate. (**b**) The Vickers hardness of O-1-to-O-5-sintered samples in the same region.

The Vickers hardness of O-1-to-O-5-sintered samples in the same region was measured and the results were presented in Figure 11b. The Vickers hardness of the O-1 sample was consistent with pure Mo [62]. When the O-2 hardness of 4500 wppm sample was 151 HV, it showed that the hardness increased slightly with an increase in O content. At this time, O can be well dissolved in the Mo matrix, and there was no obvious segregation at the GBs. The strengthening effect of O was solid solution strengthening. Compared with the O-2 sample, the hardness of the 8600 wppm O-3 sample decreased to 136 HV. The above research showed that there was a great deal of O segregation at the GBs of sintered Mo, including the presence of oxides, such as Mo dioxide and Mo trioxide.

It was well known that the O segregation at the GBs led to its weakening, making Mo more brittle. At the same time, as shown in Table 4, the growing grain size will also affect the hardness of Mo. The change in O content from O-2 to O-5 samples can also prove that it would affect Mo hardness. The minimum hardness of the O-5 sample was about 136 HV. These Mo samples with different O contents can show similar compression engineering stress–strain curves and hardness values. Since the elemental O was more segregated at the GBs of Mo, the grain boundary strengthening would be decreased, and the compressive ductility and hardness results were poor for all samples.

The grain size of sintered Mo samples with different O contents was listed in Table 4. Recrystallization and grain growth occurred during sintering. Comparing O-1 to O-5, the increase in O content from 3700 wppm to 8600 wppm made the grain size change from 17.35 μm to 24.07 μm, indicating that the existence of oxide affected the grain boundary region of Mo samples. When the O content was 8600 wppm, it was concentrated in the grain boundary regions, which slightly hindered the forward movement of the GBs, thereby inhibiting grain growth.

### 3.4. Oxygen Formation Mechanism at Grain Boundaries

In summary, we had been able to discuss the origin of O in Mo metal and its existing form in the matrix according to different characterization analysis. Fracture of the O-1-to-O-5-sintered samples was observed by SEM in Figure 6. Crystal sugar type fractures can be observed in the O-1-to-O-5-sintered samples, which were typical intergranular fractures. It was found that a large number of elemental O were distributed on GBs. The O content at the GBs was significantly higher than inside the grains in Figures 7 and 8. Meanwhile, reticular precipitates were found on the GBs of samples O-3 to O-5 in the presence of Mo grains with a precipitate width of approximately 5 μm. According to EPMA, O element, $MoO_2$, and $MoO_3$ were mainly distributed at the GBs in Figures 9 and 10. Therefore, we will discuss why these oxides were formed in Mo metal and how they were formed in GBs.

Chemical thermodynamics [63] studied the interrelationship between heat and work as well as chemical reaction or physical state change within the scope of thermodynamic laws. It determined the direction and limit of the chemical reaction and provided the final result of the reaction. They determined the thermodynamic reactions $\Delta_r G_m^\theta(T)$ at different temperatures via the thermodynamic handbook. The standard Gibbs free energy [63] expression was $\Delta_r G_m^\theta(T) = A + BT$. In order to better explore the possible reactions at different temperatures during sintering, the possible thermodynamic equilibrium reactions [64] of Mo metals in the temperature range of 0 °C to 1800 °C were calculated in Figure 12. During sintering C-O reaction or H-O reaction, possible reactions [65] include:

$$MoO_2 + 2H_2(g) = Mo + 2H_2O(g) \tag{1}$$

$$MoO_2 + 2C = Mo + 2CO(g) \tag{2}$$

$$MoO_2 + C \rightarrow Mo + CO_2 \tag{3}$$

$$Mo + O_2(g) = MoO_2 \tag{4}$$

$$2MoO_2 + O_2(g) = 2MoO_3 \tag{5}$$

$$MoO_3 + 3H_2(g) = Mo + 3H_2O(g) \tag{6}$$

$$MoO_3 + 3C = 2Mo + 3CO(g) \tag{7}$$

$$MoO_3 + 1.5C = 2Mo + 1.5CO_2(g) \tag{8}$$

$$MoO_3 + C = MoO_2 + CO(g) \tag{9}$$

$$4MoO_2 + 1.5O_2(g) = Mo_4O_{11} \tag{10}$$

$$4MoO_3 + H_2(g) = Mo_4O_{11} + H_2O(g) \tag{11}$$

$$2H_2(g) + O_2(g) = 2H_2O(g) \tag{12}$$

$$C + O_2(g) = CO_2(g) \tag{13}$$

$$2C + O_2 = 2CO(g) \tag{14}$$

We all know that, when $\Delta G$ was negative, it meant a greater probability that the reaction is happening ($\Delta$ reaction) to the right. The $MoO_2$ powder introduced into the Mo powder started to react with C in the powder after the temperature of 740 °C. The CO and $CO_2$ gas generated by the reaction was discharged from the vacuum system (Reactions (2) and (3)), as shown in Figure 12a. The $MoO_2$ in the Mo metal reacted with $H_2$ in the vacuum environment above 1084 °C to generate water vapor. The system conducted the reaction. The content of $MoO_2$ rapidly decreased $\Delta G$ and the reaction continued (Reaction (1)). During the sintering process from room temperature to 1800 °C, Mo and ambient O continued to react and produce $MoO_2$ and $MoO_3$ (Reactions (4) and (5)). Then, the product $MoO_3$ not only reacted with $H_2$ in the ambient atmosphere to form water vapor (Reaction (6)) but also reacted with the C element existing in the Mo matrix (Reactions (7) and (8)), and the obtained $MoO_3$ was also reduced by the C element to form $MoO_2$ again (Reaction (9)). As long as there was $O_2$ in the environment,

this reaction would react with Mo matrix to form $MoO_2$ and $MoO_3$ (Figure 12b). When the content of $MoO_2$ in the experiment increased to 16.4 wt.%, in addition to reactions 1–3 and 5, the introduced $MoO_2$ dioxide further reacted with the $MoO_2$ generated by the reaction to form intermediate oxide $Mo_4O_{11}$ (Reaction (10)) and intermediate oxide $Mo_4O_{11}$ (Reaction (11)) generated by reduction of $MoO_3$ via $H_2$. Since the $\Delta G$ of reaction (10) was lower than the $\Delta G$ of Reaction (11) in Figure 13b, Reaction (10) was more likely to occur before the temperature was 662 °C, and then Reaction (11) occurred. When the temperature exceeded 662 °C, introduction of excessive $MoO_2$ and the reaction-generated $MoO_2$ would continue to react with O to $Mo_4O_{11}$. In addition, the calculated results between C, H, and O would pump out the gas generated during sintering (Reactions (12)–(14)), making the thermodynamic reaction process more complete. Therefore, $MoO_2$, $MoO_3$, and intermediate oxide $Mo_4O_{11}$ phases might be generated when $MoO_2$ is introduced at a content greater than 16.4 wt.% in Mo during the sintering process. The most stable ones were $MoO_2$ and $MoO_3$, and, when the introduced $MoO_2$ content exceeded 16.4 wt.%, intermediate oxides $Mo_4O_{11}$ were generated, and then $MoO_2$ was the main reason for making $Mo_4O_{11}$ phases appear in Mo metal.

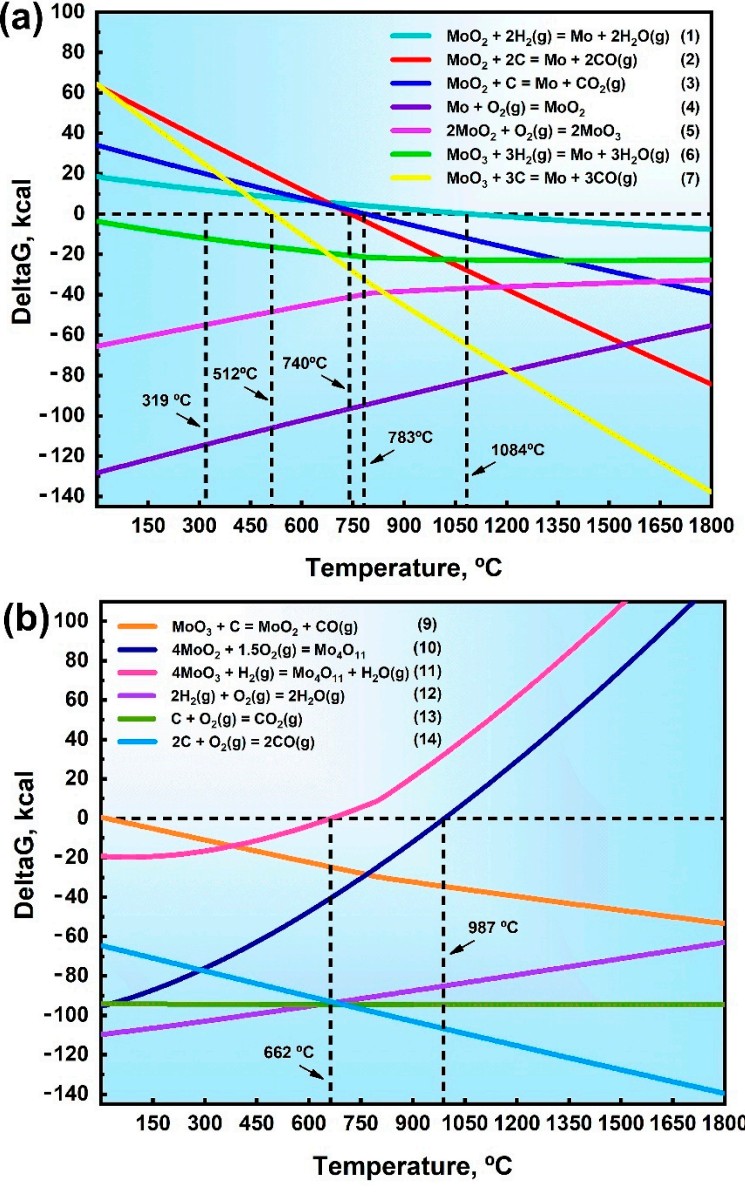

**Figure 12.** The (**a**,**b**) calculated results of thermodynamic equilibrium reactions during sintering in Mo.

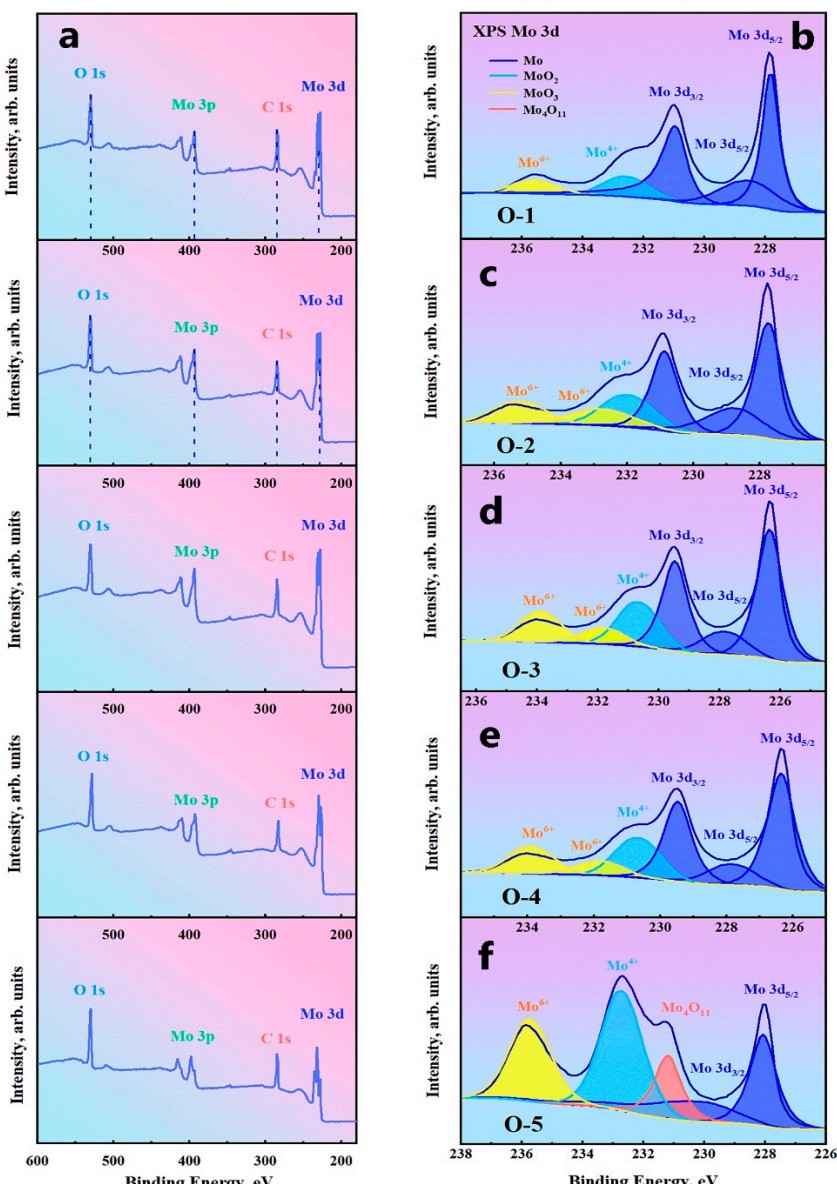

**Figure 13.** XPS spectra of O-1-to-O-5-sintered samples: (**a**) survey spectrum; (**b**–**f**) Mo 3d spectrum on the surfaces. The dark and light blue lines represent the peaks of Mo and $MoO_2$, respectively. The yellow and red lines represent the peaks of $MoO_3$ and $Mo_4O_{11}$, respectively.

In order to better verify the results of thermodynamic calculations, the oxide types of O-1-to-O-5-sintered Mo samples were qualitatively verified at the GBs by XPS. In this experiment, the type and chemical valence of the O element were qualitatively analyzed by using the binding energy. Figure 13 showed the survey spectrum and the Mo 3d spectrum on the GBs of the O-1-to-O-5-sintered samples. Further, O 1 s (530.95 eV binding energy [66]), C 1 s (284.80 eV [66]), Mo 3p (411.56 eV [66]), and Mo 3d (228 eV [66]) can be detected at the GBs of O-1-to-O-5-sintered samples in Figure 13a. The Mo 3d peaks on the grain boundary surface of the O-1-to-O-5-sintered samples with different O content were obtained by peak fitting, as shown in Figure 13b–f. It can be seen from Figure 13b–f that the Mo atom had a peak binding energy of 228 eV [66] (dark blue line). The binding energy peak of $MoO_2$ was 232.6 eV [67] (light blue line). The binding energy peaks of $MoO_3$ and $Mo_4O_{11}$ were 235.73 eV [67] (yellow line) and 231.20 eV [68] (orange line), respectively. Similarly, the results of this XPS experiment were able to verify the thermodynamic calculations, where the $O_2$ in the environment would react with Mo and the resulting $MoO_3$ would be reduced by H and C to form $H_2O$ and $MoO_2$. Eventually, the molybdenum matrix would

continue to produce $MoO_2$ and $MoO_3$, which would be detected by XPS at 1800 °C, Thus, the XPS results show that $MoO_2$, $MoO_3$, and $Mo_4O_{11}$ were contained at the GBs of the samples. The question was how these oxides form at the GBs.

Thus far, the Mo-O system was a research topic, and reliable data on the phase equilibria of Mo-O systems were inconclusive. The solubility of O in Mo was very low [69], 45 wppm and 65 wppm at 1100 °C and 1700 °C, respectively. It can be concluded that, in addition to the partial existence of O as a solid solution in Mo, free O also existed in the form of stable oxides. $MoO_2$ and $MoO_3$ were the most stable among Mo oxides, and other oxides were intermediate oxides in an unstable state. These oxides can be represented by the molecular formula $Mo_xO_{3x-1}$, and their composition was between $MoO_2$ and $MoO_3$. In this research, the causes of $MoO_2$, $MoO_3$, and $Mo_4O_{11}$ phases generation had been analyzed by thermodynamic calculations. At the same time, the oxides formed at the GBs were higher than in the grains due to easier O atoms diffusion along the GBs [70]. $MoO_3$ would also be partially volatized, and the oxidation rate would continue to accelerate. When the temperature continued to rise to 1800 °C, the O molecules existing in the vacuum would decompose into O atoms and dissolved at the GBs of Mo. According to the solubility curve, the dissolved O content was about 110 wppm. The O elements generated by the reaction react with Mo atoms to form intermediate oxides of $MoO_2$, $MoO_3$, and $Mo_4O_{11}$. Large amounts of oxides tended to aggregate at GBs and remained as the furnace cooled. Similarly, with an increase in O content, the main products were Mo and $MoO_2$ phases during sintering at 1800 °C [71] when the doped O content was less than 66 at. % in Figures 7 and 8. For the O-5 sample doped with more than 70 at. % O content, $MoO_3$ and $Mo_4O_{11}$ were able to be synthesized.

In this study, Mo with different O amount was obtained by vacuum sintering. Mo dioxide was used to control the O content in the sintered samples. In the powder metallurgy processes, the O in the powders used was mainly present on the particle surface in the form of oxides or hydroxides. The $MoO_2$ added was initially surrounded by metal during formation. O existed in the form of oxides and C existed in the green body in the form of free C. Purification and removal of impurities by vacuum sintering mainly relied on high-temperature decomposition and desorption of Mo oxides to purify the surface of Mo particles. Under the negative pressure of vacuum sintering, volatile substances were very volatile, which was conducive to decomposition of Mo oxides into Mo element, and other gaseous oxides were discharged out of systems. This allowed to advance the thermodynamic reaction process effectively so that vacuum was more effective than hydrogen sintering in degassing, deoxidation, and purification. As shown in Figure 12, the microstructure evolution behavior of O, C atoms, and Mo oxides in the Mo matrix of the O-5 sample during sintering indicated that direct and indirect carbothermal reduction reactions between $MoO_2$, $MoO_3$, and C occurred during the vacuum sintering, leading to a reduction of C and O elements in the Mo matrix. Then, degassing and deoxidation processes were the main factors affecting the O content in the sintered Mo samples [35]. Most of the doped $MoO_2$ was decomposed to obtain more carbon monoxide and water vapor for volatilization (Reactions 2, 3, and 7–9). The vacuum sintering used in this experiment can be carried out under negative pressure, which was conducive to decomposition of Mo oxides into Mo, and other gaseous oxides were discharged out of the system. During the vacuum sintering process, the CO produced by these reactions was continuously removed by the vacuum unit, and the reaction always proceeded to the right. The initial reaction temperature was about 1200–1300 °C, and the reaction was most intense between 1500 °C and 1600 °C, which could achieve the purpose of deoxidation. The C content would also decrease. Therefore, the reason for the significant reduction of O content in the five groups of sintered samples with different O contents was explained.

The mechanism diagram of microstructure evolution behavior of O, C, and Mo oxides in the Mo matrix of the O-5 sample during sintering was shown in Figure 14. Before sintering, Mo and Mo dioxide were solid–solid mixed in Figure 14a. When the sintering temperature exceeded 1800 °C, the $MoO_2$ decomposed into Mo and $MoO_3$ in Figure 14b.

It can be seen from Figure 14c that the decomposition reaction of Mo dioxide occurred, and the generated Mo trioxide was still mainly distributed at the GBs of the Mo matrix. This effectively reduced the O content of sintered Mo billet, and the O content of the five samples was in the range of 3700–8600 wppm. After furnace cooling, the $MoO_2$, $MoO_3$, and $Mo_4O_{11}$ phases would remain at GBs and distributed in the network structure (Figure 14d). This also explains why most $MoO_3$ was present in the O-5 sample because adding a large amount of $MoO_2$ would produce more $MoO_3$, thus partially retaining the grain boundary of Mo.

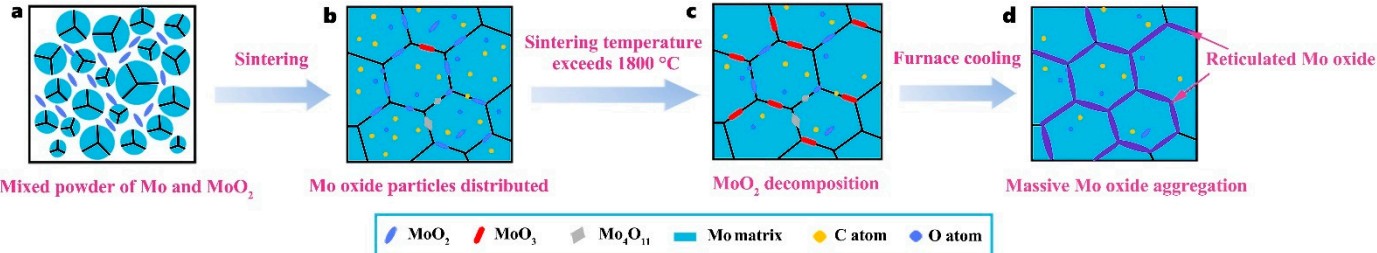

**Figure 14.** Schematic illustration of the microstructure evolution of O, C, and Mo oxides in the Mo matrix of the O-5 sample during the sintering process: (**a**) mixed powder of Mo and $MoO_2$; (**b**) distributed Mo oxide particles at GBs; (**c**) decomposition reaction of $MoO_2$; (**d**) massive Mo oxide aggregation at GBs.

## 4. Conclusions

In summary, the formation mechanism and precise composition control of interstitial O were investigated in powder metallurgy Mo. SEM, EDS, XRD, and an oxygen–nitrogen analyzer were used to examine Mo metal with different O content. The O element present in Mo metal was characterized and detected using EPMA and XPS techniques. An in-depth analysis of the sintering process at 1800 °C was carried out by thermodynamic calculations, and the authenticity of the experimental results was verified by XPS technique. The conclusions are summarized as follows:

(1) Mo samples with different O contents (3700–8600 wppm) can be prepared by adding different amounts of $MoO_2$. Under vacuum sintering conditions, the O content in the sintered samples can be controlled by changing the chemical composition design method of $MoO_2$ powder. The O element in the powder and sintering can be quantitatively detected by the oxygen–nitrogen analyzer. SEM analysis of O doping showed that the fracture mode was intergranular fracture, and the fracture morphology did not change with an increase in O content.

(2) O preferred to segregate at GBs with an increase in O content, and Mo oxides appeared in the grain boundary area of the fracture by SEM, EDS, and EPMA technology. It indicated that the solid doping process with addition of $MoO_2$ powder achieved enrichment of O at GBs and solid solution of O with Mo matrix, resulting in a maximum O element content of 8600 wppm.

(3) Thermodynamic calculations can be used as a criterion for oxide reactions in molybdenum, and the experimental results can be verified using XPS techniques. Due to addition of a large number of O elements, it was identified by EPMA and XPS that O elements in GBs exist as $MoO_2$, $MoO_3$, and intermediate oxides of $Mo_4O_{11}$ phases, and Mo oxides were distributed in GBs in a reticulated manner.

(4) The sintered O-1 had the best compressive strength of 1384.85 MPa. With an increase in O content, the compressive strength decreased gradually. Good compressive strength is demonstrated because of the solid solution strengthening effect. When the content of O increased, element O existed in the form of oxide, which led to a reduction in compressive strength, and the minimum compressive strength was 7.84 MPa. The O-4 sample with an O content of 6200 wppm had the highest yield strength (341.74 MPa). With the increase in

O contents, the hardness increased slightly from 125.86 HV to 151.92 HV and the grain size increased from 17.35 μm to 24.7 μm.

**Author Contributions:** Conceptualization, H.-R.X. and P.H.; methodology, H.-R.X.; software, C.-J.H.; validation, X.-Y.Z. and X.-J.H.; resources, K.-S.W.; investigation, J.-Y.H. and S.-W.G.; writing—original draft preparation, F.Y. and A.A.V.; visualization, W.Z. All authors have read and agreed to the published version of the manuscript.

**Funding:** This research was funded by the Outstanding Doctorate Dissertation Cultivation Fund of Xi'an University of Architecture and Technology [160842012], Scientific and Technological Innovation Team Project of Shaanxi Innovation Capability Support Plan, China [2022TD-30], the Fok Ying Tung Education Foundation [171101], Youth Innovation Team of Shaanxi Universities [2019–2022], top young talents project of "Special support program for high level talents" in Shaanxi Province [2018–2023], major scientific and technological projects in Shaanxi Province of China [2020ZDZX04-02-01], service local special program of education department of Shaanxi Province, China [21JC016], General Special Scientific Research Program of Shaanxi Provincial Department of Education [21JK0722], the General Projects of Key R&D Program of Shaanxi Province, China [2021GY-209], China Postdoctoral Science Foundation [2021M693878], and China Postdoctoral Science Foundation [2021MD703866].

**Acknowledgments:** This work was supported by the Outstanding Doctorate Dissertation Cultivation Fund of Xi'an University of Architecture and Technology (160842012), Scientific and Technological Innovation Team Project of Shaanxi Innovation Capability Support Plan, China (2022TD-30), the Fok Ying Tung Education Foundation (171101), Youth Innovation Team of Shaanxi Universities (2019–2022), top young talents project of "Special support program for high level talents" in Shaanxi Province (2018–2023), major scientific and technological projects in Shaanxi Province of China (2020ZDZX04-02-01), service local special program of education department of Shaanxi Province, China (21JC016), General Special Scientific Research Program of Shaanxi Provincial Department of Education (21JK0722), the General Projects of Key R&D Program of Shaanxi Province, China (2021GY-209), China Postdoctoral Science Foundation (2021M693878), and China Postdoctoral Science Foundation (2021MD703866). We thank Song at the Instrument Analysis Center of Xi'an University of Architecture and Technology for their assistance with the Gemini SEM 500 analysis.

**Conflicts of Interest:** The authors declare that they have no known competing financial interests or personal relationships that could have appeared to influence the work reported in this paper.

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
