# Peer review of "Exploring the Formation Mechanism, Evolution Law, and Precise Composition Control of Interstitial Oxygen in Body-Centered Cubic Mo"

_metals, doi:10.3390/met13010001_

Round 1
Reviewer 1 Report
Comments to authors
This study discusses the origin and formation of O and its effects on microstructure evolution and properties investigated in powder metallurgy Mo. The SEM, EDS, XRD, oxygen, nitrogen and hydrogen elemental analyzers were used to examine the Mo metal with different O content. This paper carefully produced with a scientific quality is good. The work was well planned and executed however there is a lack of results presentations. The kinds of literature reviewed in this manuscript and in-depth knowledge of the field are to be required. The article may be acceptable for publication after clarifying the minor revision.
The following clarification must be made to improvise the readability of the manuscript.
1. Abstract of the article is not clear and concise.
2. In the introduction section add recent literature published.
3. Mention the novelty of the adopted research in the introduction section.
4. English of the manuscript must be polished throughout the manuscript.
5. The materials and methods section will be completely revised per the journal requirement.
6. Figures quality is poor. So, provide the high-quality figures for better readability with proper legend and labels.
7. Carefully correct the typographical mistakes in the entire manuscript.
8. Results and discussion must be supported by standard literature.
9. To be provided with a detailed SEM analysis of the images, results and discussion section.
10. The conclusion is needed to write more precisely with the application of this existing methodology.
Reviewer 2 Report
Excelent manuscript. Very high scientific work. Modern methods for analyses. Thanks.
Reminders: Figure 5(c): Change Angström to nanometers.
Figure 14(b) Correct 1 atm Gas to 0.1 MPa Gas
Line 509: Correct 6.200 °C to 6,200 wppm
Reviewer 3 Report
The article is good, written in understandable language, an understandable research plan, and a clear goal, but I am curious where the carbon comes from during sintering in a vacuum. did he get it from grinding balls? please also emphasize the poor accuracy of oxygen analysis on the scanning electron microscope. Please reply.
best wishes
Reviewer
Reviewer 4 Report
This work shows an overall vision of the behaviour of intersticial impurities of oxygen in Mo structures, how analyse them and the insights of this impurity in BCC Mo metal.
The article is very well presented with many analytical and spectroscopic data. The results are connected to the data provided and explained clearly. I have no doubt that this work can be published in the present form.
